# Recent Developments in *Inonotus obliquus* (Chaga mushroom) Polysaccharides: Isolation, Structural Characteristics, Biological Activities and Application

**DOI:** 10.3390/polym13091441

**Published:** 2021-04-29

**Authors:** Yangpeng Lu, Yanan Jia, Zihan Xue, Nannan Li, Junyu Liu, Haixia Chen

**Affiliations:** Tianjin Key Laboratory for Modern Drug Delivery & High-Efficiency, School of Pharmaceutical Science and Technology, Tianjin University, Tianjin 300072, China; roylu@tju.edu.cn (Y.L.); jyn1021@126.com (Y.J.); xzh078@tju.edu.cn (Z.X.); 13234039725@163.com (N.L.); junyuliu@tju.edu.cn (J.L.)

**Keywords:** *Inonotus obliquus*, polysaccharides, hypoglycemic, anti-tumor, antiviral, applications

## Abstract

*Inonotus obliquus* (Chaga mushroom) is a kind of medicine and health food widely used by folk in China, Russia, Korea, and some occidental countries. Among the extracts from *Inonotus obliquus*, *Inonotus obliquus* polysaccharide (IOPS) is supposed to be one of the major bioactive components in *Inonotus obliquus*, which possesses antitumor, antioxidant, anti-virus, hypoglycemic, and hypolipidemic activities. In this review, the current advancements on extraction, purification, structural characteristics, and biological activities of IOPS were summarized. This review can provide significant insight into the IOPS bioactivities as their in vitro and in vivo data were summarized, and some possible mechanisms were listed. Furthermore, applications of IOPS were reviewed and discussed; IOPS might be a potential candidate for the treatment of cancers and type 2 diabetes. Besides, new perspectives for the future work of IOPS were also proposed.

## 1. Introduction

According to World Health Statistics 2020, global life expectancy increased from 66.5 years to 72.0 years during 2000 to 2016 [1]. Although the total risk of dying from any of diseases at the age of 30–70 decreased by 18%, the rate of premature deaths (defined as deaths between 30 and 70 years old) caused by diabetes has increased by 5% during this period. Recent statistics from Center for Disease Control and Prevention (CDC) has shown that 34.2 million people within the United States have diabetes till the end of 2020, which took up 10.5% of the whole population [2]. In high-income countries such as the US and EU countries, cancer has become the leading cause of premature death [1]. The American Cancer Society estimated that 1.9 million new cancer cases will be diagnosed and the number of nation-wide cancer deaths will rise to 608,570 in 2021 [3]. Beside reasonable prevention, it is also necessary and urgent to find suitable drugs to cure the disease. Unfortunately, existing chemical and biological drugs for cancers and diabetes have many drawbacks, such as large side effects and high prices [4,5,6]. Therefore, natural products that possess hypoglycemic, anti-oxidation, and anticancer properties is of great significance and drew the researchers’ attention [6,7,8,9].

As early as the 16th century, *Inonotus obliquus* was used as a folk medicine in Russia, Siberia, and some occidental countries [10,11]. It is a kind of fungus that mostly parasitize on living trunk of birch trees in the cold circumboreal region of the northern hemisphere areas [12,13]; it is also named Chaga mushroom due to its irregularly formed sterile conk with burnt charcoal like appearance [13]. Owing to abundant melanin, the sclerotium and mycelium are mostly black [14]. As most of them grow in high latitudes, the extremely low environmental temperature makes the fungus grow very slow. Thus, in the last two decades, its main active components and pharmacological research have received growing attention. As shown in Table 1, owing to various chemical components including polysaccharides, triterpenoids, polyphenols, and melanin, it has been proven to possess anticancer, anti-inflammatory, antiviral, antioxidant, hypoglycemic, and hypolipidemic activity without obvious side-effects by long-term clinical and animal experiments [15,16,17].

Polysaccharides are natural biopolymers that all living organisms have. And the diversity in monosaccharide composition and structure causes them to have different biological activities [35]. Modern research has shown that polysaccharides are involved in various bioactivities such as cell recognition, cell growth, differentiation, metabolism, embryonic development, cell cancerization, virus infection, immune response [36], etc., which are crucial for life maintenance [35]. Relevantly, it is a hot area of modern medicine and food functional chemistry research. Over several decades, remarkable progress has been realized, and especially during the past decade, polysaccharides’ anticancer, hypoglycemic, and antiviral effects were popular research topics.

Among all the active components in *Inonotus obliquus*, *Inonotus obliquus* polysaccharides have the broadest biological activity, such as anticancer, hypoglycemic, anti-inflammatory, and anti-oxidation effects [8,22,37,38,39,40,41,42,43,44,45,46,47,48,49,50,51,52,53,54,55,56,57,58,59,60,61]. Through the number of Scopus-indexed publication numbers (archived until 15 March 2021) for polysaccharide bioactivity, *Inonotus obliquus*, *Inonotus obliquus* polysaccharide related articles, a certain degree of upward trend can be seen (Figure 1), which means that the medicinal value and health value of *Inonotus obliquus* have been widely recognized and affirmed [5,6,62,63,64], making it worthwhile to review *Inonotus obliquus* polysaccharides.

In this review, the extraction, purification methods, and the physiochemical and structural features of *Inonotus obliquus* polysaccharides are summarized and discussed in detail; besides, the in vitro and in vivo evaluation and mechanisms of its bioactivities are summarized. Furthermore, the application and future perspectives are presented.

## 2. Extraction and Purification Methods

### 2.1. Extraction of IOPS

As shown in Table 2, Chen Yiyong analyzed the composition of *Inonotus obliquus* raw material; the percentage of each components are (the data were presented as means ± SD, %): moisture 3.5 ± 0.36, crude protein 2.4 ± 0.44, crude fat 1.7 ± 0.25, ash 10.4 ± 0.44, crude fiber 67.5 ± 0.95, reducing sugar 4.2 ± 0.30, polysaccharide 10.3 ± 0.40, respectively [15]. As most polysaccharides have good water solubility, the classic method uses hot water reflux to extract polysaccharides, Gao determined the optimal process for traditional water extraction by orthogonal experiments, 80 °C for temperature, 1.5 h for time, 1:40 (*w*/*v*) for material-to-liquid ratio, the yield for IOPS is 2.53% [65]. Despite that this method is simple and easy to implement, the efficiency was very low, the yield of IOPS varies from 2.12% to 2.53% [15,66].

Many people have developed modern physical methods such as microwave assist, ultrasonic heating and air technology, and cellulase-assisted extraction process to improve extraction efficiency [16,58,61,67,68,69,70,71]. Owing to the cavitation effect of ultrasonic heating, it is easier for the solvent to penetrate the cells and accelerate the dissolution of the material than in the usual condition, thus, enhancing the extraction efficacy [72]. Compared with the traditional reflux extraction, the ultrasonic assisted extraction method can not only shorten the extraction time, but also prevent the degradation of the components caused by high temperature, which is an extremely effective extraction method.

Chen used response surface analysis method to determine the optimal process parameters of ultrasonic-microwave-assisted extraction (UMAE), microwave power of 90 W, material-to-liquid ratio of 1:20 (*w*/*v*), extraction time of 19 min. Compared with the traditional water extraction method, the ultrasonic-microwave-assisted extraction time was significantly shortened while the yield of polysaccharides increased from 2.12% to 3.25% and the purity increased from 64.03% to 73.16% [15]. Dr. Xu further improved and optimized this extraction method; the optimal extraction conditions for IOPS are: ultrasonic treatment for 31 min at 52 °C temperature, with liquid-to-material ratio of 21:1 (mL/g), the IOPS extraction rate grew to 3.81 ± 0.19% [20].

Zong used the response surface analysis with four factors and three levels to obtain the optimal extraction process for IOPS: extraction by 30% concentration ethanol at 95 °C for 2.5 h, with 30:1of liquid-to-material ratio. Under this condition, the actual extraction content of IOPS was 5.993%, and the crude polysaccharide yield was 33.6% [73].

Ji used response surface optimization experiments to obtain 16.86 ± 0.48% yield for water-soluble crude polysaccharide and 28.64 ± 5.19% for alkali-soluble crude polysaccharides. The extraction temperature was 85 °C and the extraction time was 4.77 h at 43:1 ratio of liquid to material, and 26 h, with 0.6 mol/L NaOH solution while the liquid-to-material ratio was 28:1 for them [74].

An orthogonal experiment was used by Wang to determine the best process for the ultrasonic method [56]. He found the optimal conditions were an ultrasonic treatment time of 25 min, material-to-liquid ratio of 1:40, ultrasonic temperature of 50 °C, and extraction three times to obtain 8. 61% yield; a greatly improved extraction efficiency, comparing to a classical water extraction method. In addition, high-voltage pulsed electric fields and flash extraction methods have also been applied to the extraction of IOPS. The condition and yield of IOPS for different extraction methods are concluded in Table 3.

### 2.2. Purification Method of IOPS

The extracted IOPS contains impurities such as proteins, pigments, and inorganic ions in different contents, so further separation and purification are required. Currently, the traditional Seveg method and the trichloroacetic acid method are used to remove proteins [15,17,66]. Han Yaoling compared the effects of these two methods, and each has its advantages and disadvantages [75]. However, the traditional method relies on chemical reagents that usually lead to pollution. Chen used DEAE-cellulose, Sepharose CL-6B, and Sephadex G-200 column chromatography instead of the traditional method [17]. Liu developed a novel three-phase partitioning (TPP) purification technique with high efficiency and safety, and it was environmentally friendly [66]. Li compared trichloroacetic acid-n-butanol (TCA-NBA) method, hydrochloric acid method, enzymatic method, and other methods [76], among these methods, the enzymatic assisted of trichloroacetic acid-n-butanol deproteinization method presented the best results. The protein removal rate reached 82.3%, and the polysaccharide retention rate was as high as 76.1%, purity of the deproteinized polysaccharide increased to 81.1%. The activated carbon powder, hydrogen peroxide, chitosan, and polyamide chromatography column are used to decolorize the *Inonotus obliquus* polysaccharide, and a polyamide chromatography column with excellent decolorization effect was selected as the appropriate decolorization method. The decolorization rate reached 89.3% and the polysaccharide retention rate was 91.7%. The purity of the decolorized polysaccharide was improved to 85.8%.

## 3. Physiochemical and Structural Features

The structure of polysaccharides includes primary structure and advanced structure. The primary structure includes monosaccharide composition, connection sequence and the type of glycosidic bond. The high-level structures are divided into secondary, tertiary and quaternary structures. The secondary structure refers to the regular conformation formed by the hydrogen bond; the tertiary structure and the quaternary structure refer to the regular spatial conformation resulting from the non-covalent interaction between saccharide units based on secondary structure [35,77,78].

### 3.1. Monosaccharide Composition and Glycosidic Bond Connection

The structural units of polysaccharides are monosaccharides, and they are connected by glycosidic bonds. Common glycosidic bonds are α-1,4-, α-1,6-, β-1,3-, and β-1,4-glycosidic bonds. Glycosidic bonds are closely related to polysaccharide activity. The active polysaccharides obtained from bacteria are generally composed of glucose, and the β-(1, 3) -D glycosidic bond on the glucose backbone is necessary for their antitumor effects [3,4,5].

For the analysis of the primary structure of IOPS monosaccharide composition and glycoside bond connection mode, there are mainly gas chromatography (GC), periodic acid oxidation method, Smith degradation method, and methylation analysis [79].

According to Chen, the composition of IOPS were rhamnose (Rha), arabinose (Ara), xylose (Xyl), mannose (Man), glucose (Glc), galactose (Gal), with molar ratio 10.25:9.38:1:12.45:9.9:11.55. The relative molecular mass of IOPS was 156,611 Da by HPLC gel chromatography column [15]. Xue obtained two polysaccharides IOEP1 and IOEP2 from *Inonotus obliquus* liquid fermentation, IOEP1 mainly composed of galactose and mannose with 20 KDa molecular weight, while IOEP2 mainly composed of arabinose and molecular weight was 200 KDa. He found that IOEP1 and IOEP2 were all pyran-type polysaccharides, glycosidic bonds of IOEP1 and IOEP2 were both α-type and β-type via Fourier-transform infrared (FT-IR) and ^1^H-NMR spectra analysis [39].

Li et al. also studied the monosaccharide composition of IOPS by High performance liquid chromatographic (HPLC) method, he used precolumn-derivatization with 1-phenyl-3-methyl-5-pyrazolone (PMP) for the detection. The result showed that IOPS was composed of mannose, rhamnose, glucose, galactose, xylose, and arabinose with molar ratio 2.13:1.36:7.01:2.98:1:1.78 [76].

Chen used DEAE-Sepharose CL-6B column for further separation and purification to obtain four polysaccharides: IOPS1, IOPS2, IOPS3, and IOPS4 [15]. The relative molecular mass of IOPS1 was 153,172 Da and its monosaccharide composition was determined to be glucose by gas chromatography, which was presumed to be a β-glucose. The relative molecular mass of IOPS3 was determined by HPLC to be 44265 Da and its structure was analyzed by methylation analysis method, combined with GC, MS, NMR, and FT-IR methods. Wold et al. found that the glycoside bond connection were mainly: (1 → 3)-β-Glc-(1 → 3), (1 → 6)-β-Glc-(1 → 3), (1 → 6)-β-Glc-(1 → 6), (1 → 3)-β-Glc-(1 → 6), T-β-Glc-(1 → 6), (1 → 4)-β-Xyl, T-β-Xyl-(1 → 4), (1 → 6)-α-Gal-(1 → 6), (1 → 6)-α-3-O-Me-Gal-(1 → 6), (1 → 4)-α-GalA-(1 → 4), (1 → 3)-α-Man, (1 → 4)-α-GlcA and (1 → 2)-α-Rha [36].

### 3.2. High-Level Structure of Polysaccharides

There is certain correlation between spatial conformation of polysaccharides and their activities. Large number of studies have shown that polysaccharides in flexed spiral conformation have higher activity than stretchable or wrinkled ribbon-shaped polysaccharides [80,81]. Congo red experiment and UV spectroscopy showed that IOPS3 was a triple helix structure, and it was rarely influenced by temperature. However, the triple helix conformation might take place in acid, alkali, and metal ions solution [15]. Statistics has shown that triple helix conformation was the most active spatial conformation of polysaccharides. Therefore, in this way, the excellent biological activity of IOPS might contribute to the triple helix conformation [77,82].

### 3.3. Surface Morphology

Atomic force microscopy (AFM) was used to observe IOPS spatial structure and surface morphology. Figure 2 exhibits the molecular morphology of the IOPS observed under AFM. With around 0.665 nm roughness, the contact area with water molecules was increased, which showed superior anti-fatigue activity [83].

## 4. Biological Activity

### 4.1. Antitumor Activity

The anticancer mechanism of IOPS mainly includes: Decreasing the expression of MMP-2, MMP-7, and MMP-9, increasing the expression of tissue metallopeptidase inhibitor 2 (TIMP-2), and decreasing the expression of NF-κB in cancer cells [2,3,8,9,84,85,86,87,88,89,90,91].

3-(4,5-dimethylthiazol-2-yl)-2,5-diphenyltetrazolium bromide (MTT) assay and Fluorescence activated cell sorting (FACS) is commonly used for cellular cytotoxicity studies. MTT assay assesses the effects of different drugs on cell viability, and a flow cytometry experiment can further quantify the extent of cell apoptosis by Annexin-V/PI Apoptosis Detection Kit; nonviable apoptotic or necrotic cells were generally defined to be situated in double positive (PI +/Annexin-V+) quadrant [92,93].

Researchers found that IOPS have obvious antitumor effects both in vitro and in vivo, despite that some different opinions on its mechanism. Some researchers think it stimulates the immune system while the others think the antioxidative ability prevents generation of cancer cells [17,94].

Zhang used MTT assay to validate IOPS inhibition of hepatic carcinoma cell line (SMMC7721) proliferation. The result showed a concentration correlation, IOPS in the range of 1.0–16.0 μg/mL inhibition rate range from 43.6–69.2% [95].

Lee found that IOPS can effectively inhibit the migration and invasion of human non-small cell lung cancer A549 cells. Its possible mechanism might be that IOPS inhibits NF-κB nuclear translocation and JNK/AKT phosphorylation in A549 cells, and inhibits the AKT/NF-κB signaling pathway that reduces the expression level of matrix metalloproteinases (MMP); therefore, the invasion of human non-small cell lung cancer A549 cells was inhibited [96,97].

Zhang found that IOPS can induce apoptosis of ovarian cancer cell SKOV3. He speculated that the mechanism might be that IOPS reduces the mRNA expression of Bcl-xl gene by promoting the expression of p53 gene mRNA, thereby regulating the process of cell apoptosis [49].

Li verified IOPS inhibition effect on human cervical cancer cells (Hela), mouse bone marrow tumor cells (SP2), and human Chang’s liver cells (Chang) through the MTT method [76]. Compared with 5-fluorouracil, the antitumor effect of IOPS was significant, and it was a promising antitumor substance. At a range of 10–1000 μg/mL IOPS, proliferation on Hela, SP2, and Chang cells were obviously inhibited. The growth inhibition rates were from 4.78% to 46.54%, 2.89% to 44.90%, 3.56% to 22.56%, respectively. And it showed a concentration-dependent relationship. There was a significant difference in the growth inhibition rate between each concentration (*p* < 0.01).

Chen also used the MTT method to study the inhibitory effect of IOPS on Jurkat and Daudi tumor cells in vitro, and the results showed that IOPS had a significant in vitro inhibitory effect on the proliferation of tumor cells Jurkat and Daudi (*p* < 0.01), and it had dose-dependent, and its maximum inhibition rates reached 71.84% and 75.14%, respectively [15]. Besides, he established a Jurkat tumor-bearing nude mice model to study in vivo effect of IOPS. The results showed that given a daily dose of 50–100 mg/kg IOPS for 10 days, the tumor inhibition rate varied from 43.52% to 57.48%. He also studied the release of Cytc and the expression of caspase-3 enzyme via Western blotting. By the aid of DNA ladders gel electrophoresis analysis, the mechanism of IOP3a-induced tumor cell apoptosis was determined: by stimulating tumor cell mitochondria to release Cytc, caspase-3 was activated and endogenous was activated. Endonuclease caused a DNA cleavage to achieve its antitumor effect [98].

Lin Yan studied the antitumor activity of IOPS by constructing a Kunming mouse S180 tumor-bearing model, and found that the inhibition rate was positively correlated with the IOPS dose. Among them, the inhibition rate of *Inonotus obliquus* intracellular and extracellular polysaccharide on S180 were 30.13–41.07%, 23.29–36.73%, respectively. In addition, compared with the control group, *Inonotus obliquus* extracellular polysaccharide could significantly increase the thymus index and spleen index of S180 tumor-bearing mice (immune indicators) [99].

Besides, Jiang’s research demonstrated that IOPS could significantly inhibit allograft tumor growth of the LLC1 cells with increased apoptosis at 50 mg/kg dosage [100]. Li tested the efficacy of IOPS by using a mouse model of Colitis-Associated Cancer (CAC) induced by azoxymethane and dextran sulfate sodium (AOM/DSS) to confirm that IOPS inhibited the proliferation of SW620 colorectal cancer cells [22].

IOPS on human osteosarcoma MG-63 cells and U2OS cells’ s proliferation, invasion, migration, and apoptosis was studied in vitro by CCK-8 assays, cell scratch assays, trans-well assays, and flow cytometry. Moreover, 320 μg/mL IOPS significantly inhibited proliferation, decreased migration and invasion ability, and increased apoptosis rate (*p* < 0.05) as IOPS could significantly inhibit the activation of the Akt/mTOR and NF-κB signaling pathway (*p* < 0.05) [101].

Yang obtained C-OWC contained with about 70% *Inonotus obliquus* polysaccharide by water extraction and ethanol precipitation. It could increase the mRNA expression of Bax, P21 and Bid genes in human liver cancer cells HepG2, while significantly reducing the expression of Bcl-2 genes. The mechanism of action might be that C-OWC increased the expression of P21 gene mRNA, and blocked the cell cycle of HepG2 cells in the G0/G1 phase, thereby inhibiting cell division and proliferation. Meanwhile it increased the mRNA expression of Bax and Bid genes in cells and reduces Bcl- 2 gene mRNA expression, thereby inducing HepG2 cell apoptosis [102]. The diagram of possible mechanisms is shown in Figure 3 and they are concluded in Table 4.

### 4.2. Hypoglycemic and Hypolipidemic Activities

The main mechanism of IOPS hypolipidemic activities may be attributed to the increase of lecithin-cholesterol acyl transferase (LCAT) activity, which promotes the synthesis of high-density lipoprotein cholesterol (HDL-C2), and transfers the free cholesterol in peripheral tissues to the liver, catalyzes the hydrolysis of triglycerides (TG) into glycerol and fatty acids, and promotes the conversion of high-density lipoprotein 3-cholesterol (HDL3-C) into HDL-C2, promotes the reverse transport and metabolism of cholesterol, thereby reducing serum total cholesterol (TC) content [104]. The hypoglycemic activity of polysaccharides is closely related to its structure. It could be affected by monosaccharide composition, molecular weight, branched chain structure, high-level structure, etc. Existing studies have found that the mechanisms of polysaccharides for hypoglycemic activity mainly include regulation of enzyme activity, liver glucose metabolism, intestinal flora, protection and reparation of pancreatic islet cells, and improvement of insulin sensitivity [105,106,107,108,109].

Wang and Zhang investigated the anti-diabetic effects and the potential mechanism of IOPS in vivo and found that IOPS ameliorated insulin resistance and lipid metabolism disorders in streptozotocin-induced type 2 diabetic mice. The possible hypoglycemic and hypolipidemic mechanism might supposedly be via the regulation of the PI3K/Akt and AMPK/ACC signal pathways [40,64]. As Wang’ s results had shown, in comparison with controlled diabetic mice, 900 mg/kg oral administration of IOPS reduced 28% serum glucose in mice, significantly restored the body and fat weight, and lowered fasting blood glucose levels, improved glucose tolerance ability and, thus, increased the level of hepatic glycogen and improved insulin resistance. Zhang found that given *I. obliquus* extract at 500 mg/kg orally, the blood glucose and insulin resistance of diabetic mice significantly alleviated while the cholesterol transportation in the liver enhanced, liver glycogen content and high-density lipoprotein cholesterol (HDL-C) levels increased with significant decline in total cholesterol (TC), triglyceride (TG) and low-density lipoprotein cholesterol (LDL-C) levels. Zhang supposed that the upregulated protein expression levels of phosphatidylinositol-3 kinase (PI3K), p-protein kinase B (Akt), p-adenosine monophosphate activated protein kinase (AMPK) and p-acetyl-CoA carboxylase (ACC) with downregulating in fatty acid synthase (FAS) and sterol regulatory element-binding protein-1c (SREBP-1c) also contributed to the hypoglycemic effects of IOPS [64].

α-Amylase and α-glucosidase are enzymes that promote the breaking of α-1,4-glycosidic bonds in starch, maltose, sucrose and oligosaccharides and hydrolyze them into monosaccharide that are easy to digest and be absorbed by the human body, such as glucose [106]. This caused the postprandial blood glucose levels of diabetic patients to increase and insulin sensitivity to be reduced, even lead them to serious complications and aggravate the condition [110]. Chen et al. found that IOPS could reduce the absorption of carbohydrates by inhibiting the activity of intestinal α-amylase and α-glucose, effectively reducing postprandial blood glucose levels, repairing islet damage, and improving complications [111]. Cong Wang investigated the simulated digestion of *Inonotus obliquus* polysaccharide UIOPS-1 in vitro. The digested polysaccharide (UIOPS-1I) after digestion could inhibit α-amylase up to 37.96% at a concentration of 200 μg/mL, which was significantly higher than the initial state of UIOPS-1 (*p* < 0.001). He speculated that it might be caused by the changes in monosaccharide composition and the decrease in molecular weight [38,40,54,55].

Through in vitro experiments, Xue found that IOEP1 and IOEP2 strongly increased HepG2 cells’ glucose consumption and insulin-resistant HepG2 cells, which proved that IOEP1 and IOEP2 might be suitable anti-diabetes agents in functional foods and natural drugs [39]. With significant anti-diabetic effects, IOPS might be a promising candidate for the clinical treatment of diabetes mellitus [112]. The possible hypoglycemic mechanism of IOPS are shown in Figure 4.

### 4.3. Antioxidant Activity

The human body continuously produces various reactive oxygen free radicals during the oxidative metabolism process of life activities, which may easily cause oxidative damage to DNA and further cause cancer, arteriosclerosis, aging, and other diseases. Polysaccharides and polysaccharide metal ion complexes have excellent antioxidant activity, and their antioxidant mechanisms mainly include: direct capture or removal of active oxygen such as OH, O_2_, and H_2_O_2_ inhibit the progress of the oxidation reaction, thereby blocking the initiation of the lipid peroxidation reaction; complexes of metal ions such as Fe^2+^ and Cu^2+^ that are necessary for the production of active oxygen inhibit the production of active oxygen; promote the production of SOD, CAT, GSH-Px and other antioxidant enzymes release or increase their activity, which indirectly exerts antioxidant effects [20,113,114,115,116,117].

IOPS antioxidant activities were frequently investigated by hydroxyl radical assay, superoxide radical assay, and ferric-reducing antioxidant power assay. Raw 264.7 cell models are often built to assess the in vitro activity [9,10,59,69,117,118,119,120]. The results showed that IOPS exhibited antioxidant activities, and the higher the content of uronic acid and proteinous substances, the stronger the antioxidant activities of polysaccharides [121,122,123,124,125,126]. Besides, molecular weights of polysaccharides also influence their antioxidant activities. IOP3a and IOP4 showed higher antioxidant properties than IOP1b, IOP2a, and IOP2c [126].

Sim’s study revealed that IOPS protection against H_2_O_2_-induced oxidative damage in RINm5F pancreatic β cells at 25–100 μg/mL [127]. Chen found that 1 mg/mL IOPS scavenged 49.72% hydroxyl radical and IC_50_ for DPPH scavenging was 0.62 ± 0.01 μg/mL [120,128]. IC_50_ of superoxide anion scavenging was 425 μg/mL [122].

Mu investigated IOPS antioxidant properties via reactive species (ROS), 1,1’-Diphenyl-2-picrylhydrazyl (DPPH) radical, hydroxyl radical and superoxide anion radical, as well as their protective effects on H_2_O_2_-induced PC12 cell death. Results indicated that IOPS could scavenge all ROS and rescue PC12 cell viability from 38.6% to 79.8% and 83.0% at a concentration of 20 μg/mL [125].

### 4.4. Anti-Fatigue Activity

The mice swimming model is frequently the test model [129]. Zhong used 64 male mice and Zhang used 40 Kunming mice (four weeks old) for an anti-fatigue activity study. Four groups were given polysaccharide fractions 1 of *Inonotus obliquus* (PIO-1), polysaccharide fractions 2 of *Inonotus obliquus* (PIO-2), polysaccharide fractions 3 of *Inonotus obliquus* (PIO-3) and saline (control group) orally for 30 days, and mice were submitted to the forced-swimming test [83,129]. Total duration of climbing, swimming, and immobility in the last 5 min of the 6 min test session, blood lactic acid (BLA), urea nitrogen (BUN), and lactic dehydrogenase (LDH) were collected for analysis. Total integrated optical density (IOD) that represents the 5-HT expression levels in brains was determined. As Figure 5 and Figure 6 have shown, the forced swimming test showed that the 50 mg/kg dose of PIO-1 could increase the climbing time and swimming time and reduce the immobile time, which indicated that PIO-1 reduced blood lactic acid (BLA), urea nitrogen (BUN), and lactate dehydrogenase (LDH) levels and significantly reduced the 5-HT concentration (total integrated optical density (IOD) representing the expression levels) in the mouse brain.

Reproduced with permission [83].

### 4.5. Antiviral Effect

A virus is a submicroscopic infectious agent that infects all types of life forms, from animals and plants to microorganisms, including bacteria and archaea [130]. The virus has brought great harm to human society; especially the Covid-19 virus that broke out in Wuhan in 2019 and has caused 2,681,790 deaths as of 18 March 2021, according to Center for Systems Science and Engineering (CSSE) at Johns Hopkins University (JHU) [131]. Therefore, the research on antiviral drugs is of great significance.

Currently, there are relatively few studies on the antiviral mechanism of *Inonotus obliquus*, and further research is needed. It was reported that IOPS could inhibit the induction of NO and other similar cytokines, a phenomenon that had been associated with Covid-19 [132,133]. By using cell models of feline calicivirus (FCV) in vitro, Tian revealed the mechanism of IOPS treatment was that it might induce its inhibitory actions directly on virus particles through blocking viral binding/absorbing, and this demonstrated that IOPS might be a potential broad-spectrum antiviral drug against feline viruses [134]. Pan investigated the antiviral mechanisms of AEIO (aqueous extract from *Inonotus obliquus,* contains IOPS), and unlike nucleoside analog antiherpetics, it could effectively prevent the entry of HSV-1 by acting on viral glycoproteins and lead to the prevention of membrane fusion [135].

According to Ichimura et al., a high-molecular-weight water-soluble lignin derivative extracted from *Inonotus obliquus* by boiling water extraction had significant inhibitory activity on HIV-1 protease. He speculated that the possible mechanism might be that this lignin derivative was adsorbed on the protease and inhibited the reverse transcriptase of HIV, thus, achieving the effect of inhibiting the reproduction of HIV. In contrast, those low-molecular-weight polyphenols and low-molecular-weight lignin could not inhibit the protease [136].

### 4.6. Immunomodulatory Activity

Fan found that polysaccharide ISP2a significantly enhanced the lymphocyte proliferation and increased the production of TNF-α, demonstrating IOPS to be a potential application as natural antitumor agent with immunomodulatory activity [36,46,137].

Won investigated the immune stimulating activity of IOPS and the signaling pathway of IOPS-mediated macrophage activation in RAW264.7 macrophage cells [137]. It was found that IOPS effectively promoted macrophage activation through the MAPK and NF-κB signaling pathways and potentially regulated the immune response, as IOPS promoted NO/ROS production, secreted more TNF-α and uptook phagocytic in macrophages, cell proliferation, comitogenic effect and IFN-γ/IL-4 secretion in mouse splenocytes.

The activation of NLRP3 inflammasome can cause the overexpression and secretion of inflammation-related cytokines, and mediate the occurrence of inflammation. When the activation of NLRP3 inflammasome is inhibited, the inflammation-related diseases involved in it may be improved. In Lipopolysaccharide (LPS)-induced macrophage inflammation, Zheng found that IOPS can inhibit the expression of IL-18 and IL-1β by inhibiting the phosphorylation of NF-κB-p65 (*p* < 0.05) [22,138,139].

By IOPS oral administration, in vivo growth of melanoma tumor was suppressed in tumor-bearing mice. Harikrishnan revealed that IOPS could positively enhance the innate immune system that effectively promote the health status of kelp grouper against *V. harveyi* infection [140]. The possible anti-inflammation mechanism is shown in Figure 7.

### 4.7. Other Bioactivities

Hu revealed that IOPS could regulate gut microbiota of chronic pancreatitis in mice [141]. Chen studied IOPS’ anti-inflammatory effect, and the results showed that IOPS alleviated inflammatory responses by inhibiting JAK-STAT signaling pathways that regulate the release of T Helper subsets, and could alleviate dextran sodium sulfate-induced chronic murine intestinal inflammation in mice at 100 mg/kg [142].

*Toxoplasma gondii* (*T. gondii*) infection caused liver injury model in mice are frequently applied for hepatoprotective effects. Xu studied IOPS hepatoprotective effects and mechanism in vivo via this model [143]. The results showed that IOPS treatment significantly decreased the liver coefficient, alanine aminotransferase (ALT) level, aspartate aminotransferase (AST), malondialdehyde (MDA) and nitric oxide (NO), and increased the contents of antioxidant enzyme superoxide dismutase (SOD) and glutathione (GSH). The protection was partly due to its anti-inflammatory effects through inhibiting the TLRs/NF-κB signaling axis and the activation of an antioxidant response by inducing the Nrf2/HO-1 signaling.

Chen et al. optimized the extraction process and obtained IOPS composed with mannose, glucose, galactose, and xylose of 1.00:4.49:1.00:0.85 mass ratio. Comparing with traditional extraction methods, their patent has shown the modified preparation method: washing the water extract with absolute ethanol and acetone three times and adding 5:1 (*v/v*) chloroform-n-butanol solution, followed by dialyzing with 3500 Da molecular weight dialysis bag in deionized water for 48 h, and then the collected dialysate was freeze-dried. The obtained IOPS could treat atopic dermatitis, food allergy, allergic rhinitis or asthma, especially for systemic allergies caused by OVA in animal experiments; thus, IOPS could be developed into an anti-allergic drug [144].

## 5. Toxicity of IOPS

Despite that most polysaccharides are low-toxic with no significant side effects, it is still necessary to evaluate the potential risks and toxicity of polysaccharides in detail before clinical research [104]. Usually, the toxicity is studied via MTT assay or an animal model. Zheng studied the cytotoxicity of IOPS toward Raw264.7 cells via MTT assay. Different concentrations of IOPS and PBS (control group) were added to a 96-well plate, and RAW264.7 cells were incubated for 24 h; there were no significant difference between the control group and the IOPS group in cell survival rate. The results also showed that 20–160 μg/mL of IOPS administration was safe and would not produce drug toxicity [138]. Dr. Gao studied the in vitro cytotoxicity of *Inonotus obliquus* ethyl acetate fraction towards normal hepatic cells. The results showed that *Inonotus obliquus* ethyl acetate was nontoxic to L02 cells and could protect L02 cells from oxidative damage caused by hydrogen peroxide. However, the ethyl acetate fraction only contained part of IOPS, not pure IOPS [145]. Cong Wang studied the sub-acute toxicity of *Inonotus obliquus* polysaccharides-chromium (III) complex in normal mice. His result confirmed that high dose administration of IOPS had neither an obvious influence on serum profiles levels nor an antioxidant ability, as the organ tissues of the toxicity group maintained organization and integrity after having been given a high dose of IOPS chromium (III) complex [146]. Briefly, 20 mice were randomly separated into two groups; the normal control group (NC) received normal saline, while the toxicity group (TC) received IOPS chromium (III) complex at a daily dose of 1500 mg/kg. All the mice were sacrificed after four weeks’ administration, and the liver, pancreases, kidney, heart, thymus, and spleen were collected and weighted, and then the liver, pancreases, and kidney were fixed with 10% formalin for histological analysis. Both these results suggested that IOPS had the potential to be a good candidate for functional food or even pharmaceuticals agents.

## 6. Application

For centuries, *Inonotus obliquus* (Chaga mushroom) has been used in Kiev to cure lip tumor, and it is a traditional medicine in Siberia [147]. Nowadays, a growing number of *Inonotus obliquus* related functional foods are developed. As Liu’s patent CN104366193-A has shown, healthcare food composed of IOPS will be used for preventing and treating diabetes, reducing blood fat, removing free radicals, protecting the heart, and regulating immune function.

Owing to its wide range of pharmacological activities, including hypoglycemic activity, improving immunity, anti-fatigue activity, and various beneficial effects, IOPS is used in functional and health products in Russia, China, and the US. Reviewing the existing products on e-commerce platforms such as Amazon and Taobao, we found that most commercial applications are health products, tea, and tincture. The existing and potential IOPS applications are listed in Table 5 and Table 6. Some healthy food products include Sorlife Chaga capsules, IOPS tea, tablets, and Chaga briquettes, which are listed on e-commerce platforms, where they were called natural insulin. To improve the quality demands of customers and enhancing the manufacturer’s reputation, the manufacturer might implement ISO22000:2018 and the HACCP methodology [148].

## 7. Conclusions and Future Perspectives

As more and more patients tend to find alternative therapies for cancer and diabetes, the awareness towards seeking natural products has increased in both academia and in society. IOPS exhibited enormous potential in immunity enhancement, and antitumor, antioxidant, anti-fatigue, hypoglycemic, and hypolipidemic activity, providing an alternative way for treatment of cancer and diabetes. The present review of *Inonotus obliquus* polysaccharides mainly focused on the recent advances in the isolation, purification, physiochemical characterization and bioactivities. By applying new extraction method, the yield of IOPS can be significantly increased. IOPS exhibited significant hypoglycemic, hypolipidemic, antioxidant, and anti-fatigue activity, and cytotoxicity toward multiple cancer cells, including hepatic carcinoma cells, human non-small cell lung cancer cells, ovarian cancer cells, human cervical cancer cells, etc. The low-toxic property and no obvious side effects make it more attractive.

Although the development of IOPS has achieved advanced progress in the last decade, some problems still cannot be ignored. Due to the variations of habitat environment and extraction methods, the composition and content of the obtained polysaccharides are not the same. Thus, the establishment of planting and extraction standard is of great importance. Furthermore, despite that more and more remarkable pharmaco-logical advances are being made, the present commercial applications are still limited to several functional foods. On the one hand, deep investigations in precise mechanisms underlying their bioactivities and the structure-activity relationships (SAR) are crucial. On the other hand, further investigations that focus on toxicity and potential risks are also to be conducted to provide solid scientific and theoretical guidance for synthesizing IOPS derivatives with enhanced biological activity and less side effects. Meanwhile, the development of related products, especially commercial applications are also encouraged, in that more and more people can benefit from the multiple bioactivities of natural polysaccharides.

## Figures and Tables

**Figure 1 polymers-13-01441-f001:**
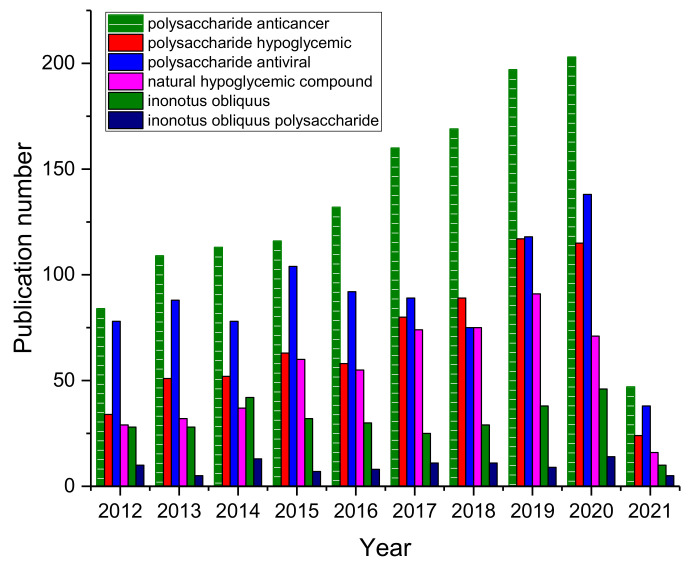
Scopus-indexed publication numbers for polysaccharide bioactivity, *Inonotus obliquus*, *Inonotus obliquus* polysaccharide related articles. (Archived until 15 March 2021).

**Figure 2 polymers-13-01441-f002:**
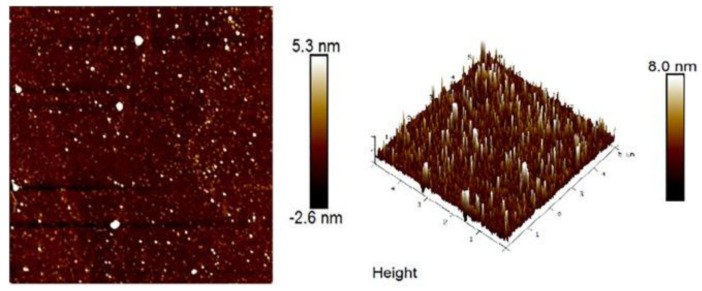
AFM analysis of IOPS. Reproduced with permission [83].

**Figure 3 polymers-13-01441-f003:**
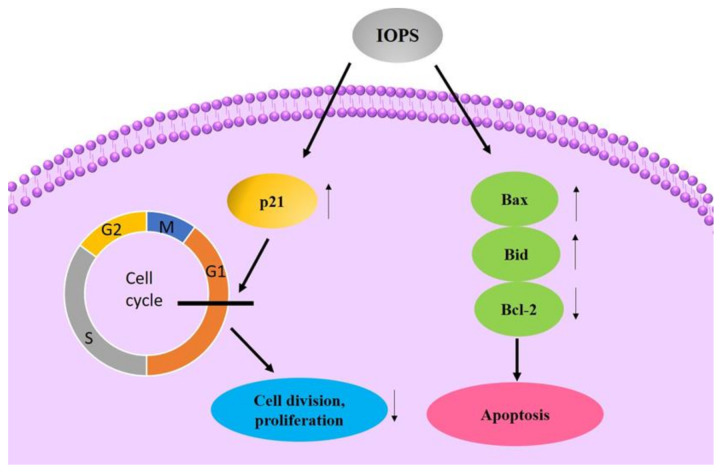
Possible anticancer mechanism of IOPS.

**Figure 4 polymers-13-01441-f004:**
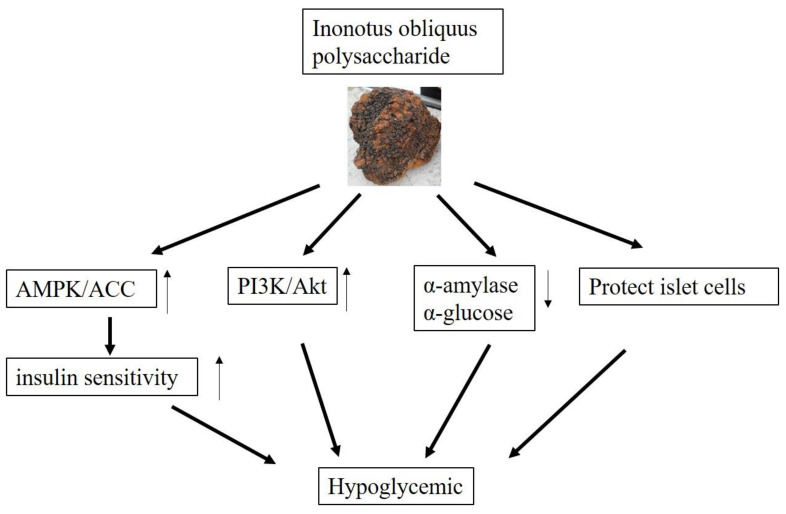
Possible hypoglycemic mechanism of IOPS.

**Figure 5 polymers-13-01441-f005:**
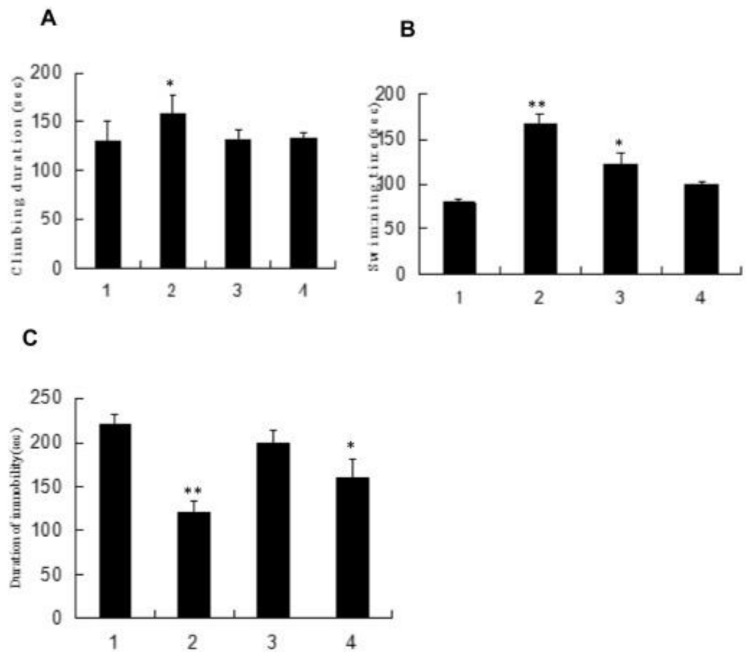
Effect of polysaccharide fractions on mice force swimming test. (**A**) Climbing duration of mice. (**B**) Swimming time of mice. (**C**) Duration of immobility. All bars represent mean values with vertical lines indicating S.E.M. Number of animals = 10. * *p* < 0.05 vs. control group. ** *p* < 0.01 vs. control group (1—Saline-treated group; 2—PIO-1-treated group; 3—PIO-2-treated group; and 4-PIO-3-treated group).

**Figure 6 polymers-13-01441-f006:**
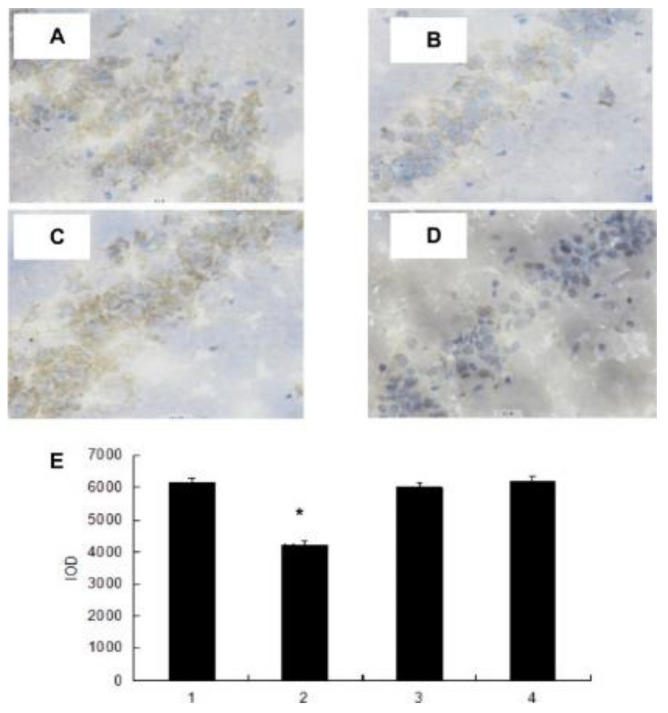
Effect of polysaccharide fractions on 5-HT concentrations in mice. Histological examination on 5-HT concentrations in mice (**A**) Saline-treated group; (**B**) PIO-1 treated group; (**C**) PIO-2 treated group; and (**D**) PIO-3 treated group) H&E staining results; (**E**) all bars represent mean values with vertical lines indicating S.E.M. Number of animals = 10. * *p* < 0.01 vs. control group (1—Saline-treated group; 2—PIO-1 treated group; 3—PIO-2 treated group; and 4—PIO-3 treated group). Reproduced with permission [83].

**Figure 7 polymers-13-01441-f007:**
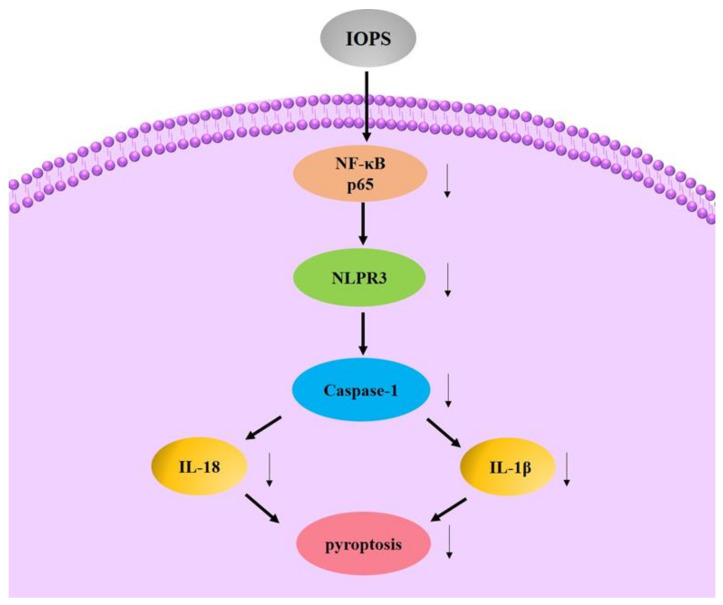
Possible anti-inflammation mechanism of IOPS.

**Table 1 polymers-13-01441-t001:** Biological activities of *Inonotus obliquus* main components.

Component	Biological Activities	References
Polysaccharide	Anticancer, anti-inflammatory, antiviral, antioxidant, immunomodulatory, hypoglycemic, hypolipidemic, hepatoprotective, etc.	[18,19,20,21,22,23,24]
Polyphenol	Antioxidant	[25]
Triterpenoid	Anticancer, anti-inflammatory, antiviral, and antioxidant	[26,27,28,29,30]
Melanin	Antioxidant, anti-inflammatory, antiviral, hypolipidemic and immunomodulatory	[24,31,32,33,34]

**Table 2 polymers-13-01441-t002:** Composition of *Inonotus obliquus* raw material.

Composition	Percentage Content (Mean ± SD, %)
Moisture	3.5 ± 0.36
Crude protein	2.4 ± 0.44
Crude fat	1.7 ± 0.25
Ash	10.4 ± 0.44
Crude fiber	67.5 ± 0.95
Reducing sugar	4.2 ± 0.30
Polysaccharide	10.3 ± 0.40

**Table 3 polymers-13-01441-t003:** Condition and yield of IOPS for different extraction method.

Extraction Method	Extracting Temperature (°C)	Extracting Time	Solid/Water Ratio (*w*/*v*)	Polysaccharide Yield (%)	References
Water Extraction	80	1.5 h	1:40	2.53	[65]
Water Extraction	85	4.77 h	1:43	16.86 ± 0.48	[74]
UMAE		19 min	1:20	3.25	[15]
UMAE	52	31 min		3.81 ± 0.19	[20]
30% Ethanol Extraction	95	2.5 h	1:30	5993	[73]
0.6 mol/L NaOH		26 h	1:28	28.64 ± 5.19 (alkali-soluble crude polysaccharides)	[74]
Ultrasonic Method	50	25 min	1:40		[56]

**Table 4 polymers-13-01441-t004:** Anticancer studies of IOPS.

Cancer/Tumor Type	Concentration/Dose	Inhibition Rate	Mechanism	References
Hepatic carcinoma cell line (SMMC7721)	1.0–16.0 μg/mL	43.6–69.2%	N/A	[95]
Human non-small cell lung cancer cells (A549)	100 μg/mL	26%	NF-κB nuclear translocation (−), JNK/AKT phosphorylation (−), AKT/NF -κB (−), MMP expression level (−), Invasion (−)	[45,96,97]
Ovarian cancer cells (SKOV3)	160 μg/mL	23.94%	p53 gene mRNA expression (+), Bcl-xl gene mRNA expression (−), regulating cell apoptosis	[49]
Human cervical cancer cells (Hela)	10–1000 μg/mL	4.78–46.54%	N/A	[76]
Mouse bone marrow tumor cells (SP2)	10–1000 μg/mL	2.89–44.90%	N/A	[76]
Human Chang’s liver cells (Chang)	10–1000 μg/mL	3.56–22.56%	N/A	[76]
Human T lymphocytic leukemia cells (Jurkat)	200 μg/mL 50–100 mg/kg·d	71.84% 43.52–57.48%	Tumor cell mitochondria release Cytc(+), caspase-3 (+), endogenous (+), DNA cleavage	[15,98]
Human B lymphocyte tumor cells (Daudi)	200 μg/mL	75.14%	N/A	[15]
Kunming mouse S180 tumor	200 mg/kg·d	41.07% (intracellular polysaccharide) 36.73% (extracellular polysaccharide)	N/A	[99]
Human cervical cancer cells (Hela)	200 μg/mL	57.7%	N/A	[103]
Osteosarcoma cells (MG-63 and U2OS)	320 μg/mL	22.3% (MG-63) 23.64% (U2OS)	Akt/mTOR (−), NF-κB (−)	[101]

(+): improve or promote; (−): inhibit or reduce. N/A: information was not available.

**Table 5 polymers-13-01441-t005:** Existing IOPS application.

Application	Function	Brand
Health product	Hypoglycemic	Komsomlski
Health product	Enhance immunity	Wonder Land Herbs
Health product	Hypoglycemic	Sorlife
Tea	Enhance immunity	Johncan production
Tea	Enhance immunity, antioxidant	Atomic nature
Tincture		Florida Herbs

**Table 6 polymers-13-01441-t006:** Potential IOPS application.

Application	Function	References
Beverage	Hypoglycemic	[149]
Healthcare food	Preventing and treating diabetes	[150]
Functional yogurt	Enhance immunity, hypoglycemic	[151]
Drug	Anti-allergic	[144]

## Data Availability

Data are contained within the article.

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
