# Peer review of "Recent Developments in Inonotus obliquus (Chaga mushroom) Polysaccharides: Isolation, Structural Characteristics, Biological Activities and Application"

_polymers, 2021, doi:10.3390/polym13091441_

Round 1

Reviewer 1 Report

This paper reports on the “Recent developments in Inonotus obliquus (Chaga mushroom) polysaccharides: Isolation, structural characteristics, biological activities and application”. The article is interesting and clearly presented. Particularly noteworthy is the presentation of both the scientific side and industrial application and future opportunities. Reference seems to be correct.

I have few comments to the manuscript:

  1. All manuscript. Quote before the end of the sentence and not superscript eg products [1].
  2. In the introduction, there are no literature references to some of the statements.
  3. The paragraphs do not contain short summaries from the authors, which may make the article seem inconsistent in some parts.

Taking into account all comments the manuscript may be published in Polymers after minor revision.

Reviewer 2 Report

In general, the manuscript of " Recent developments in Inonotus obliquus (Chaga mushroom) polysaccharides: Isolation, structural characteristics, biological activities and application " needs slight revise to reach the level of Polymers. The comments and problems are as follows:

  1. In the first paragraph of introduction, the authors used a lot of numbers and percentages to prove the current situation of diabetes. Please add references to supporting the information.
  2. In the last sentence of the first paragraph of introduction, please add references to supporting this type of information.
  3. The introduction section is too short. Please consider increase this section.
  4. Please carefully check the formatting and punctuation of the article, such as (W/V) should change into (w/v).
  5. In Biological activity, the authors focused too much on the result and ignored to the principle which polysaccharides can exert biological activity. Please add this part of content in the manuscript.
  6. The conclusion should mainly reflect the research results and their theoretical value, practical value, present research situation and prospect. It is suggested to modify the conclusion.
  7. Please refer to the papers of recent ten years, unless the old references are from reputable journals.

Author Response

Response to Reviewer 2 Comments

Journal: Polymers

Manuscript ID:  polymers-1169629

Title: Recent developments in Inonotus obliquus (Chaga mushroom) polysaccharides: Isolation, structural characteristics, biological activities and application

Dear Editors and Reviewers,

Thank you very much for your useful comments and suggestions on our manuscript, which are very important to improve the quality and significance of this manuscript. Herein, we clarify all the revisions and explanations in our manuscript as following.

Point 1:

In the first paragraph of introduction, the authors used a lot of numbers and percentages to prove the current situation of diabetes. Please add references to supporting the information.

Response: Thank you very much for your great suggestion. We add the references to the statement including the statistic numbers from Center for Disease Control and Prevention (CDC), American Cancer Society and WHO.

Point 2: In the last sentence of the first paragraph of introduction, please add references to supporting this type of information.

Response: Thank you very much for your great suggestion. We had added the references.

Point 3: The introduction section is too short. Please consider increase this section.

Response: Thank you very much for your great suggestion. We have increased this section.

Point 4: Please carefully check the formatting and punctuation of the article, such as (W/V) should change into (w/v).

Response: Thank you very much for your great suggestion. We have checked the formatting and punctuation, including w/v, in vitro and in vivo.

Point 5: In Biological activity, the authors focused too much on the result and ignored to the principle which polysaccharides can exert biological activity. Please add this part of content in the manuscript.

Response: Thank you very much for your great suggestion. We have summarized the mechanism of anticancer, hypoglycemic and hypolipidemic activities, and add them to the Biological activity chapter.

Point 6: The conclusion should mainly reflect the research results and their theoretical value, practical value, present research situation and prospect. It is suggested to modify the conclusion.

Response: Thank you very much for your great suggestion. We have rewritten the conclusion.

Point 7: Please refer to the papers of recent ten years, unless the old references are from reputable journals.

Response: Thank you very much for your great suggestion. We tried to add 13 more new references to replace the old ones, but some of the old references did not have suitable replacements, so we did not delete them completely.

We tried our best to improve the manuscript and made some changes in the manuscript.  We appreciate for your warm work earnestly, and hope that the correction will meet with approval.

Once again, thank you very much for your suggestions. Looking forward to hearing from you.

Thank you and best regards.

Yours sincerely,

Haixia Chen

Reviewer 3 Report

Certainly, the authors present a topic that can be interesting and with an attractive and innovative title. However, the manuscript has serious deficiencies that make it difficult to consider its acceptance in this important journal. A thorough modification is very necessary.
-The authors must focus the study topic much more adequately, justifying the importance of microorganisms and their beneficial effects.
-Applicability should be the central point at first.
-The importance of the new organic formulations must go a second time, accompanied by their own diagrams that represent the concept that is intended to be transmitted.
-Figure 3 and 4 are not understood in the context, they must be represented in a more adequate way justifying the presence of proteins. Authors must justify more sophisticated studies.
-The images should be more self-explanatory and with a more transversal character for the reader.
-Table 2 is very deficient, the MTT technique is understood when one justifies and explains the content of the table.
-Authors must include existing trademarks and patents. The copyright of all displayed images must be specifically mentioned in the manuscript. In this sense, the authors should include the already existing translation of these formulations.
-Toxicity studies must be specifically and appropriately included in the review.
-The applicability should be extended much more extensively.
-The authors must create explanatory figures that transmit the concepts.
-Authors should check the English grammar of their manuscript.

Author Response

Response to Reviewer 3 Comments

Journal: Polymers

Manuscript ID:  polymers-1169629

Title: Recent developments in Inonotus obliquus (Chaga mushroom) polysaccharides: Isolation, structural characteristics, biological activities and application

Dear Editors and Reviewers,

Thank you very much for your useful comments and suggestions on our manuscript, which are very important to improve the quality and significance of this manuscript. Herein, we clarify all the revisions and explanations in our manuscript as following.

Point 1:

The authors must focus the study topic much more adequately, justifying the importance of microorganisms and their beneficial effects.

Response: Thank you very much for your great suggestion. We have rewritten the introduction and add the mechanism, application of its biological activities to justify its importance.

Point 2: Applicability should be the central point at first.

Response: Thank you very much for your great suggestion. We had added the application of this fungi. However, most applications are limited to health food or tea.

Point 3: The importance of the new organic formulations must go a second time, accompanied by their own diagrams that represent the concept that is intended to be transmitted.

Response: Thank you very much for your great suggestion.  However, we did not fully understand the meaning of new organic formulations, and it seems that it could hardly be found in literature.

Point 4: Figure 3 and 4 are not understood in the context, they must be represented in a more adequate way justifying the presence of proteins. Authors must justify more sophisticated studies.

Response: Thank you very much for your great suggestion. We have replaced the Figure 3 and 4 with text to justify its anticancer mechanism. The anti-cancer mechanism of IOPS mainly includes: decreasing the expression of MMP-2, MMP-7 and MMP-9, increasing the expression of tissue metallopeptidase inhibitor 2 (TIMP-2) and decreasing the expression of NF-κB in cancer cells

Point 5: The images should be more self-explanatory and with a more transversal character for the reader.

Response: Thank you very much for your great suggestion. We have deleted the inappropriate Figure 3 and 4, and replaced by text descriptions.

Point 6: Table 2 is very deficient, the MTT technique is understood when one justifies and explains the content of the table.

Response: Thank you very much for your great suggestion. We have deleted the column.

Point 7: Authors must include existing trademarks and patents. The copyright of all displayed images must be specifically mentioned in the manuscript. In this sense, the authors should include the already existing translation of these formulations.

Response: Thank you very much for your great suggestion. We have deleted the images of commodities, and add Table 4 to summarize existed trademarks.

Point 8: Toxicity studies must be specifically and appropriately included in the review.

Response: Thank you very much for your great suggestion. We have added the Toxicity of IOPS.

Point 9: The applicability should be extended much more extensively.

Response: Thank you very much for your great suggestion. IOPS have good biological activities both in vitro and in vivo, however, most applications are limited to health food or tea, no clinical trial has been reported.

Point 10: The authors must create explanatory figures that transmit the concepts.

Response: Thank you very much for your great suggestion. We have added the graphical abstract to explain the concept.

Point 11: Authors should check the English grammar of their manuscript.

Response: Thank you very much for your great suggestion. We have checked the grammar of the manuscript.

We appreciate for your warm work earnestly, and hope that the correction will meet with approval.

Once again, thank you very much for your suggestions. Looking forward to hearing from you.

Thank you and best regards.

Yours sincerely,

Haixia Chen

Round 2

Reviewer 3 Report

The authors have not adequately made the suggested changes. The authors carry out a non-exhaustive review, providing data without adequate structuring. The figures and graphs provided by the authors do not have the quality that a review of these characteristics should have. The text does not provide relevant data to be cited by other authors.

Author Response

Response to Reviewer 3 Comments

Journal: Polymers

Manuscript ID:  polymers-1169629

Title: Recent developments in Inonotus obliquus (Chaga mushroom) polysaccharides: Isolation, structural characteristics, biological activities and application

Dear Editors and Reviewers,

Thank you very much for your useful comments and suggestions on our manuscript, which are very important to improve the quality and significance of this manuscript. Herein, we clarify all the revisions (highlighted in RED) and explanations in our manuscript as following.

Round 2: Point 1: The authors have not adequately made the suggested changes. The authors carry out a non-exhaustive review, providing data without adequate structuring. The figures and graphs provided by the authors do not have the quality that a review of these characteristics should have. The text does not provide relevant data to be cited by other authors.

Response: Thank you very much for your great suggestion. We have revised the manuscript according to your suggestion. We have summarized the mechanism of anticancer, hypoglycemic and anti-inflammation activities, and drew 3 diagrams to illustrate the mechanisms. Secondly, we explained the polysaccharide structure, compositions, including structure activity relationship. Thirdly, we improve the information and organization.

Round 1: Point 1:

The authors must focus the study topic much more adequately, justifying the importance of microorganisms and their beneficial effects.

Response: Thank you very much for your great suggestion. We have rewritten the relevant background of cancer and diabetes to prove the significance of the study. The mechanisms of IOPS are also added, besides, more potential applications are added.

Point 2: Applicability should be the central point at first.

Response: Thank you very much for your great suggestion. We had added the existing and potential application of this fungi.

Point 3: The importance of the new organic formulations must go a second time, accompanied by their own diagrams that represent the concept that is intended to be transmitted.

Response: Thank you very much for your great suggestion.  We highlighted the importance of the new organic formulations, and drew 3 new diagrams (Fig 3,4,7) to represent the concept.

Point 6: Table 2 is very deficient, the MTT technique is understood when one justifies and explains the content of the table.

Response: Thank you very much for your great suggestion. We have deleted the column and added the possible mechanisms.

Point 8: Toxicity studies must be specifically and appropriately included in the review.

Response: Thank you very much for your great suggestion. We have added 2 toxicity study of IOPS.

Point 11: Authors should check the English grammar of their manuscript.

Response: Thank you very much for your great suggestion. We have re-checked the grammar of the manuscript.

We tried our best to improve the manuscript and made some changes in the manuscript. We appreciate for your warm work earnestly, and hope that the correction will meet with approval.

Once again, thank you very much for your suggestions. Looking forward to hearing from you.

Thank you and best regards.

Yours sincerely,

Haixia Chen
